# Improved Representation of Asymmetrical Distances with Interval Quasimetric Embeddings

**Tongzhou Wang**                                          TONGZHOU@MIT.EDU
**Phillip Isola**                                              PHILLIPI@MIT.EDU
*MIT CSAIL*

**Editors:** Sophia Sanborn, Christian Shewmake, Simone Azeglio, Arianna Di Bernardo, Nina Miolane

## Abstract

Asymmetrical distance structures (quasimetrics) are ubiquitous in our lives and are gaining more attention in machine learning applications. Imposing such quasimetric structures in model representations has been shown to improve many tasks, including reinforcement learning (RL) and causal relation learning. In this work, we present four desirable properties in such quasimetric models, and show how prior works fail at them. We propose Interval Quasimetric Embedding (IQE), which is designed to satisfy all four criteria. On three quasimetric learning experiments, IQEs show strong approximation and generalization abilities, leading to better performance and improved efficiency over prior methods.
**Keywords:** Quasimetrics, Asymmetry, Representation Geometry, Representation Learning, Reinforcement Learning

## 1. Introduction

We live in a geometric world. As we move our arms along smooth curves in the 3-dimensional Euclidean space, or find short paths w.r.t. the Manhattan(-like) distance of a city, we are interacting with one of the essential geometric artifacts—distance. Distances are at the core of almost all decision making.

Although commonly modelled as symmetric quantities, distances and costs are rarely reversible. Wind, gravity, and magnetic forces naturally make one direction harder than the other. Human-designed rules, such as one-way roads, form another source of asymmetry. Fundamental quantities, time and entropy, are inherently irreversible. Decision making in this *asymmetrical* world generally is based on *asymmetrical distances*, called *Quasimetrics* (Wang and Isola, 2022; Mémoli et al., 2018).

Quasimetrics capture the essence of comparing options and optimal planning—the triangle inequality, without requiring symmetry. It is a natural structure for many machine learning problems. In reinforcement learning and control, optimal goal-reaching plan costs in Markov decision processes are exactly quasimetrics (Bertsekas and Tsitsiklis, 1991; Tian et al., 2020; Pitis et al., 2020). In casual inference, causal relations can also be formulated as quasimetrics (Balashankar and Subramanian, 2021). Directed graph embedding and (hierarchical) relation discovery are also special cases of learning quasimetrics (Vendrov et al., 2015; Ganea et al., 2018; Suzuki et al., 2019). In such tasks, many work have demonstrated the benefit of imposing a quasimetric structure in model representations (Wang and Isola, 2022; Liu et al., 2022; Balashankar and Subramanian, 2021; Venkattaramanujam et al., 2019).

Table 1: Various quasimetric modeling methods on the four criteria from Section 2 **(last four columns)**. Methods are grouped by their inherent latent (quasi)metric structure **(first column)**. ‡PQE reparametrizes its few effective parameters (scale factors) via deep linear networks with many parameters, to overcome optimization difficulties (likely due to diminishing gradients from not being positive homogeneous) †Deep Norm and Wide Norm were previously believed to not universally approximate (Wang and Isola, 2022; Liu et al., 2022), but we show that they in fact do in Section 3.1. §Deep Norm and Wide Norm by default use learned concave activations to transform its components, which are not positive homogeneous (Figure 1). *MRN as described in the original paper is not a quasimetric but can be easily modified to be one. We compare with both versions in experiments.

| Latent Structure | Method | Quasimetric Constraints | Universal Approx. | Latent Quasimetric Head #Parameters | Latent Positive Homogeneity |
|---|---|---|---|---|---|
| No Latent Quasimetric | Unconstrained Quasimetric Predictors (e.g., Tian et al., 2020; Rizi et al., 2018; Nair et al., 2018) | ✗ | ✓ | — | ✗ |
|  | Dot Product of Asymmetrical Encoders (e.g., Schaul et al., 2015; Hong et al., 2021) | ✗ | ✓ | None | ✗ |
| Latent Metric | Metric Embeddings | ✓ | ✗ | Usually None | Usually ✓ |
| Latent Quasimetric | Poisson Quasimetric Embedding (PQE) (Wang and Isola, 2022) | ✓ | ✓ | A Lot (for optimization reasons)‡ | ✗ |
|  | Deep Norm (Pitis et al., 2020) | ✓ | ✓† | A Lot | ✗§ |
|  | Wide Norm (Pitis et al., 2020) | ✓ | ✓† | A Lot | ✗§ |
|  | Metric Residual Network (MRN) (Liu et al., 2022) | ✗* | ✓ | A Lot | ✗ |
|  | **Interval Quasimetric Embedding (This paper)** | ✓ | ✓ | None | ✓ |

In this paper, we consider different ways to add such latent quasimetric structures to model representations. Section 2 discusses four desired properties: (1) satisfying quasimetric constraints, (2) universal approximation, (3) low predictor parameter count, and (4) latent positive homogeneity. On a high-level, (1) and (2) ensure correct geometric inductive bias and coverage, while (3) and (4) are related to better optimization and easier usage in downstream models and/or layers.

To our best knowledge, no prior method satisfies all four requirements. Section 3 introduces Interval Quasimetric Embedding (IQE) as a new quasimetric embedding approach that fulfills all criteria. In Section 5, we empirically verify that IQE significantly improves over previous methods on three quasimetric learning tasks from Wang and Isola (2022).

## 2. Latent Quasimetric Structures

**Definition 1 (Quasimetric)**  *Given set $\mathcal{X}$ and function $d\colon \mathcal{X} \times \mathcal{X} \to [0,\infty]$, $(\mathcal{X},d)$ is a quasimetric space if $d$ satisfies*

- *(triangle inequality) $\forall x,y,z,\ d(x,z) \leq d(x,y) + d(y,z)$;*
- *(identity) $\forall x,\ d(x,x) = 0$.*

There are many approaches to impose latent quasimetric structures (Definition 1) in machine learning models. Since such models essentially capture certain quasimetric distances in data, we call them *quasimetric models*.

One may simply formulate a quasimetric model as an unconstrained predictor network that approximates certain quasimetrics (Schaul et al., 2015; Nair et al., 2018; Rizi et al., 2018). However, the learned model may not fully respect quasimetric constraints. In fact, it is proved that they may violate these constraints arbitrarily badly (Wang and Isola, 2022).

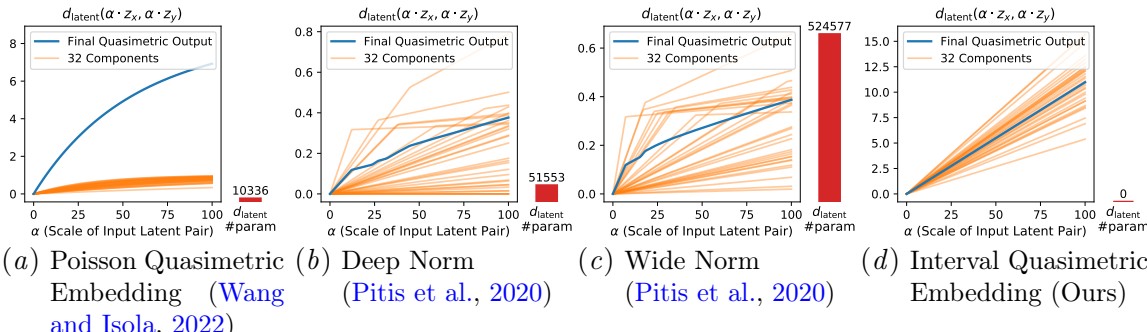

(a) Poisson Quasimetric Embedding (Wang and Isola, 2022)  (b) Deep Norm (Pitis et al., 2020)  (c) Wide Norm (Pitis et al., 2020)  (d) Interval Quasimetric Embedding (Ours)

Figure 1: Different latent quasimetrics $d_{\mathsf{latent}}$. Plots show how predicted distances (and components forming them) change as two latent vectors move apart. Red bars show the number of trainable parameters in $d_{\mathsf{latent}}$. (a) PQE suffers from diminishing gradients. (b,c) Deep Norm and Wide Norm require expensive latent quasimetric head, and have complex relations between latents and predictions (due to its learned concave transformations). (d) IQE uses a simple head and does not suffer from gradient optimization issues. (a-d) Plots are computed at random initializations, with Deep Norm and Wide concave transformation parameters scaled to emphasize the non-linearity.

Alternative approaches employ a *latent quasimetric* function on a learned latent space (Pitis et al., 2020; Wang and Isola, 2022). They always respect the quasimetric properties by construction, but can vary in terms of expressivity, efficiency, and effectiveness for using in downstream models.

We evaluate different methods based on four criteria. First two are basic requirements for a general and proper quasimetric structure, and directly relate to the generalization capabilities (Theorem 4.3 of Wang and Isola (2022)): the formulation must be able to represent **all quasimetrics** and (ideally) **only quasimetrics**:

1. **(Quasimetric Constraints)** Quasimetric model should (at least approximately) satisfy all quasimetric properties, enforcing the correct geometry and inductive biases.

2. **(Universal Approximation)** Quasimetric model should be able to generally approximate any quasimetric structure of data (with jointly trained encoder models).

To our best knowledge, only certain *latent quasimetric* formulations satisfy both above properties. These methods jointly learn an *encoder* $f(x; \theta)$ mapping data $x \in \mathcal{X}$ to a latent vector $z_x \in \mathcal{Z}$, as well as a (sometimes parametrized) *latent quasimetric head* $d_{\mathsf{latent}}(z_x, z_y; \psi)$ estimating the distance from data $x$ to data $y$ (via their latents). Both components collectively define a parametrized quasimetric on data space $\mathcal{X}$:

$$d(x, y; \theta, \psi) \triangleq d_{\mathsf{latent}}(f(x; \theta), f(y; \theta); \psi), \qquad x \in \mathcal{X}, y \in \mathcal{X}. \quad (1)$$

General Latent Quasimetrics

Not all such formulation are equally good at optimization and obtaining a high-quality latent space for transferring to downstream models and/or tasks. We argue that $d_{\mathsf{latent}}$ should be a simple mapping that satisfies two criteria (visualized in Figure 1):

3. **(Latent Quasimetric Head $d_{\mathsf{latent}}$ with Few Parameters)** A mapping with many trainable parameters complicates the relation between input latent and quasimetric

distances. This not only hurts training efficiency but also makes it harder for subsequent models to take advantage of the quasimetric inductive bias in latent vectors. Instead, most parameters and processing should ideally be in the encoder $f$.

4. **(Latent Positive Homogeneity:** $d_{\mathsf{latent}}(\alpha z_x, \alpha z_y) = \alpha \cdot d_{\mathsf{latent}}(z_x, z_y)$**,** $\forall \alpha > 0, z_x, z_y$**)** This is similar to requiring the gradient gradient of $d_{\mathsf{latent}}$ (along certain directions) to have small Lipschitz constant, a property known to improve speed and stability of gradient optimization (Sashank et al., 2018; Li and Orabona, 2019). As shown in Figure 1(a), this is not even approximately true in the prior work—Poisson Quasimetric Embedding (PQE), leading to to diminishing gradient and a limited output range. Indeed, PQE requires complex reparametrization tricks to aid optimization (Wang and Isola, 2022). This property linearly relates quasimetric distances with latent magnitudes (with directions fixed), and potentially allows downstream deep models (that are good at processing Euclidean inputs) to better utilize the quasimetric structure.

Table 1 summarizes prior works and our proposal on all four properties. Next section describes our proposed Interval Quasimetric Embedding (IQE), the only method that satisfies all four requirements.

## 3. Interval Quasimetric Embeddings (IQE)

The main issue with PQE is that its components are bounded in $[0, 1)$ and suffer from diminishing gradients (Figure 1(a)). Special reparametrization tricks are necessary for successful optimization (Wang and Isola, 2022). We propose Interval Quasimetric Embeddings (IQE) to directly address this drawback. Appendix A derives IQE via a modified PQE framework.

IQE is a new encoder-based quasimetric model, where a (learned) encoder maps data into some latent space, where our latent IQE quasimetric $d_{\mathsf{IQE}}$ outputs a quasimetric distance between two given latents.

**IQE Components.** Similar to PQE, IQE considers input latents as two-dimensional matrices (via reshaping). For input latents $u, v \in \mathbb{R}^{k \times l}$, IQE is formed by components that capture the total size (i.e., Lebesgue measure) of unions of several intervals on the real line:

$$\forall i = 1, 2, \ldots, k, \qquad d_i(u, v) \triangleq \underbrace{\left| \bigcup_{j=1}^{l} \underbrace{\left[ u_{ij}, \max(u_{ij}, v_{ij}) \right]}_{\text{interval on the real line}} \right|}_{\text{size of the set formed from union of } l \text{ intervals}}. \qquad (2)$$

$$\text{IQE Components}$$

Figure 2 provides a graphical illustration on how to compute these components.

**Combining IQE Components.** Unlike PQE, IQE components are positive homogeneous and can be arbitrarily scaled (Figure 1(d)), and thus do not require special reparametrization in combining them. Simply summing yields the most basic yet effective IQE formulation, IQE-sum:

$$d_{\mathsf{IQE\text{-}sum}}(u, v) \triangleq \sum_{i=1}^{k} d_i(u, v) \qquad (3)$$

$$\text{IQE-sum}$$

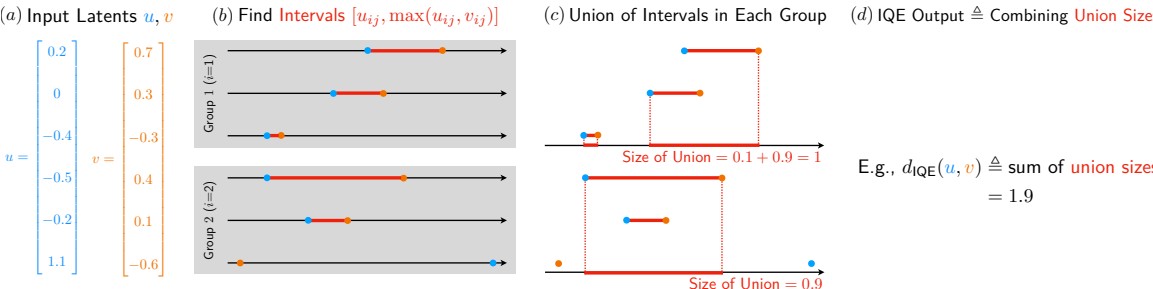

Figure 2: Computing IQE quasimetric from latent $u \in \mathbb{R}^{2 \times 3}$ to latent $v \in \mathbb{R}^{2 \times 3}$.

Using the maxmean reduction from prior work (Pitis et al., 2020), we obtain IQE-maxmean with a single extra parameter $\alpha \in [0, 1]$ (parametrized via a `sigmoid` transform):

$$d_{\mathsf{IQE\text{-}maxmean}}(u, v; \alpha) \triangleq \mathrm{maxmean}(d_1(u, v), \ldots, d_k(u, v); \alpha) \qquad (4)$$

IQE-maxmean

$$\triangleq \alpha \cdot \max(d_1(u, v), \ldots, d_k(u, v))$$
$$+ (1 - \alpha) \cdot \mathrm{mean}(d_1(u, v), \ldots, d_k(u, v))$$

Prior methods often require expensive predictor heads (e.g., MRN, Deep Norm and Wide Norm) and/or complex initialization and reparametrization (e.g., PQEs). In contrast, both IQE formulations have very simple forms. In Section 5, we will see that IQEs are not only simple, but also practically effective.

### 3.1. Theoretical Results on Universal Approximation

Following prior works (Wang and Isola, 2022; Liu et al., 2022; Pitis et al., 2020), we assume that the target quasimetric $(\mathcal{X}, d)$ has only finite distances. Here we present strong universal approximation gurantees for IQEs. All full proofs are in Appendix B.

**Theorem 2 (IQE Universal Approximation; Finite Case)** *For any finite quasimetric space $(\mathcal{X}, d)$ with $|\mathcal{X}| = n < \infty$, there exists encoders $f_1, f_2$ such that $(f_1, d_{\mathsf{IQE\text{-}maxmean}})$ exactly represents $d$, and $(f_2, d_{\mathsf{IQE\text{-}sum}})$ approximates $d$ with distortion $\mathcal{O}(t \log^2 n)$, where $t$ is a complexity measure of $(\mathcal{X}, d)$ (called treewidth).*

**Proof** [Sketch] IQE-maxmean can exactly represent function $d_{\mathsf{asym}}(u, v) = \max_i (v_i - u_i)^+$. Rewriting $d(x, y) = \max_{z \in \mathcal{X}}(d(x, z) - d(y, z))^+$ leads a desired encoder $f_1$.

For IQE-sum, each IQE component can exactly represent any quasimetric that takes in binary values (called quasipartitions) with arbitrary scaling. The desired distortion can be achieved with a convex combination of quasipartitions (Lemma C.5 of Wang and Isola (2022)), and thus also with IQE-sum. ∎

**Theorem 3 (IQE Universal Approximation; General Case)** *Consider any quasimetric space $(\mathcal{X}, d)$ where $\mathcal{X}$ is compact and $d$ is continuous. $\forall \epsilon > 0$, with sufficiently large $m$, there exists some continuous encoder $f \colon \mathcal{X} \to \mathbb{R}^m$ such that*

$$\forall x \in \mathcal{X}, y \in \mathcal{X}, \quad |d_{\mathsf{IQE\text{-}maxmean}}(f(x), f(y)) - d(x, y)| \leq \epsilon. \qquad (5)$$

**Relation with PQE.** IQE-maxmean guarantees are strictly stronger than those of PQEs (and IQE-sum), which is only a distortion bound on the finite case using polynomial-sized encoders. With the same encoder, IQE-maxmean exactly represents any finite quasimetric.

**Relation with MRN.** Our IQE-maxmean analysis is largely inspired by the MRN results. In Appendix B, full proofs reduce the MRN asymmetrical component to an IQE-maxmean.

**Deep Norm and Wide Norm.** Also using a connection to MRN, we are the first to prove that Deep Norm and Wide Norm universally approximate any *quasimetric*.

**Theorem 4 (Deep Norm and Wide Norm Universal Approximation)** *Deep Norm and Wide Norm enjoy the same approximation results stated in Theorems 2 and 3.*

## 4. Related Works

**Quasimetric** captures a common geometric structure. Perhaps the most important example is the cost of navigating between any two states, which is extensively studied in control theory and reinforcement learning (RL) (Bertsekas and Tsitsiklis, 1991). Such costs directly correspond to Q-functions and value functions (Schaul et al., 2015), making quasimetric models an increasingly popular choice for them (Pitis et al., 2020; Tian et al., 2020; Wang and Isola, 2022; Liu et al., 2022). In relation learning, causality (Balashankar and Subramanian, 2021) and hierarchy (Vendrov et al., 2015; Ganea et al., 2018) discovery can be modelled as learning (special cases of) quasimetrics, and benefit from quasimetric models.

**Latent Quasimetrics and Representation Learning.** Latent quasimetric is the most common approach to model quasimetrics. A (usually jointly learned) encoder maps data into a latent space, where some (maybe parametrized) latent quasimetric function gives the distance prediction output (Pitis et al., 2020; Wang and Isola, 2022; Liu et al., 2022). In theoretical computer science, manually constructed quasimetric embeddings are used in an improved sparse-cut algorithm (Mémoli et al., 2018). In machine learning, latent quasimetrics can be used to obtain representation spaces that are directly informative of quasimetric structures (Balashankar and Subramanian, 2021; Vendrov et al., 2015). Indeed, representation learning methods are often designed to capture certain geometric properties, including similarity (Wang and Isola, 2020), and equivalences (Zhang et al., 2020; Wang et al., 2022), and independence/disentanglement (Burgess et al., 2018).

## 5. Experiments

IQE satisfies all four requirements from Section 2. But does it perform better empirically?

We use all three tasks from Wang and Isola (2022) to evaluate (1) the ability to approximate quasimetrics and generalize to test pairs (2) benefits from enforcing a quasimetric structure in deep learning models. For quasimetric modeling tasks, accurate prediction on test pairs (i.e., generalization) is directly related to good approximation of training distances and respecting quasimetric constraints (Wang and Isola, 2022).

For all tasks, we compare the following eight families of quasimetric models:

- **IQEs (Ours):** IQE-sum and IQE-maxmean.
- **PQEs (Wang and Isola, 2022):** PQE-LH and PQE-GG.

Table 2: Modeling the large-scale Berkeley-Stanford Web Graph with different quasimetric models. For some **baseline** families, we show the best method picked w.r.t. validation set MSE.

| | | Validation Set Metrics | | |
|---|---|---|---|---|
| | | MSE w.r.t. $\gamma$-discounted distances ($\times 10^{-3}$) $\downarrow$ | $\ell_1$ error when true $d < \infty$ $\downarrow$ | Predicted distance when true $d = \infty$ $\uparrow$ |
| IQE-sum | | **1.078** $\pm$**0.053** | **1.303** $\pm$**0.031** | 118.244 $\pm$5.412 |
| IQE-maxmean | | 1.488 $\pm$0.307 | 1.333 $\pm$0.218 | 89.635 $\pm$1.726 |
| PQE-LH | | 2.921 $\pm$0.187 | 1.659 $\pm$0.048 | 71.390 $\pm$0.436 |
| PQE-GG | | 3.872 $\pm$0.136 | 2.121 $\pm$0.146 | $\infty$ (overflow) |
| Wide Norm | | 3.533 $\pm$0.212 | 1.769 $\pm$0.021 | 124.658 $\pm$2.868 |
| Deep Norm | (Original) | 5.071 $\pm$0.135 | 2.085 $\pm$0.063 | 120.045 $\pm$4.353 |
| | (+ Non-Negativity Fix) | 4.760 $\pm$0.354 | 2.035 $\pm$0.057 | 120.151 $\pm$4.700 |
| MRN | (Original) | 10.820 $\pm$0.817 | 2.882 $\pm$0.205 | 129.528 $\pm$4.237 |
| | (+ Quasimetric Fix) | 6.875 $\pm$0.333 | 2.508 $\pm$0.091 | 129.914 $\pm$6.291 |
| Best Metric Embedding | | 17.595 $\pm$0.267 | 7.540 $\pm$0.074 | 53.850 $\pm$3.843 |
| Best Unconstrained Net. | (No Regularizer) | 3.086 $\pm$0.039 | 2.115 $\pm$0.024 | 59.524 $\pm$0.370 |
| | (+ $\Delta$-Ineq. Regularizer) | 2.813 $\pm$0.063 | 2.211 $\pm$0.034 | 61.371 $\pm$0.394 |
| Best Asym. Dot Product | (No Regularizer) | 48.106 $\pm$0.006 | 2.520 $\times 10^{11}$ $\pm$2.175 $\times 10^{11}$ | 2.679 $\times 10^{11}$ $\pm$2.540 $\times 10^{11}$ |
| | (+ $\Delta$-Ineq. Regularizer) | 48.102 $\pm$0.000 | 2.299 $\times 10^{11}$ $\pm$9.197 $\times 10^{10}$ | 2.500 $\times 10^{11}$ $\pm$1.446 $\times 10^{11}$ |

- **Unconstrained Neural Networks:** Unconstrained networks that map concatenated input pair to a raw distance prediction (directly, with exp transform, and with $(\cdot)^2$ transform) or $\gamma$-discounted distance (directly, and with a sigmoid-transform).

- **Asymmetrical Dot Products:** On input pair $(x, y)$, encoding each into a feature vector with a *different* network and then taking the dot product. Identical to unconstrained networks, the output is used in the same 5 ways.

- **Metric Encoders:** Embedding into Euclidean space, $\ell_1$ space, hypersphere with (scaled) spherical distance, or a mixture of all three with learned weights.

- **Deep Norm (Pitis et al., 2020):** The original formulation may produce negative values. We use both the original version and a fixed version (see Appendix C.1).

- **Wide Norm (Pitis et al., 2020).**

- **MRN (Liu et al., 2022):** The original formulation may violate quasimetric constraints. We use both the original version and a fixed version (see Appendix C.2).

**Unconstrained Models and A Triangle Inequality Regularizer.** Unlike other methods that explicitly enforce quasimetric structures, both unconstrained networks and and asymmetrical dot products can represent any function, and are unconstrained models. We evaluate these methods because (1) that they are widely used (Tian et al., 2020; Hong et al., 2021; Rizi et al., 2018; Schaul et al., 2015) and (2) that their performances reveal whether standard training of generic models can somehow implicitly learn the underlying quasimetric structure in data. Additionally, we test whether explicit regularization can help these unconstrained models better learn quasimetric structure, and train them with a triangle inequality regularizer $\mathbb{E}_{x,y,z}\left[\max(0, \gamma^{\hat{d}(x,y)+\hat{d}(y,z)} - \gamma^{\hat{d}(x,z)})^2\right]$ for weights $\in \{0.3, 1, 3\}$.

All results are aggregated from 5 seeds. Full details are provided in Appendix D.

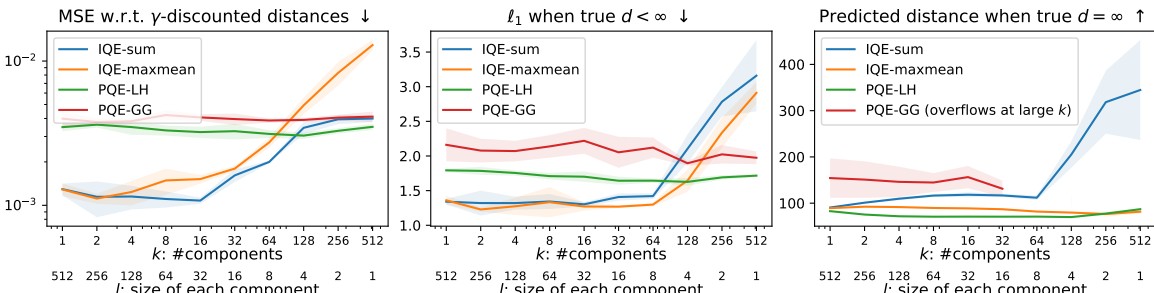

Figure 3: Effect of different $(k, l)$ choices for IQEs and PQEs with fixed total latent dimension $= 512$.

### 5.1. Large-Scale Social Graph

Berkeley-Stanford Web Graph (Leskovec and Krevl, 2014) is a large real-wold social graph, containing 685,230 pages as nodes, and 7,600,595 hyperlinks as directed edges. Following prior work (Wang and Isola, 2022), we use 128-dimensional node2vec features (Grover and Leskovec, 2016) and the landmark method (Rizi et al., 2018) to obtain a training set of 2,500,000 pairs, and a validation set of 150,000 pairs.

**IQEs significantly improve modeling large real-world graphs.** We train various quasimetric models to approximate the training distances by minimizng MSE w.r.t. $\gamma$-discounted distance with $\gamma = 0.9$. In Table 2, both IQEs greatly outperform all baselines, attaining lowest MSE, accurately predicting finite distances, and outputting high predictions for infinite (unreachable) pairs. Compared to the prior best methods, the simple IQE-sum has a 61% improvement on MSE and a 16% improvement on $\ell_1$ error (on finite distances).

**Effects of $k$ and $l$.** Both IQE and PQE interpret input latent vectors as two-dimensional $\in \mathbb{R}^{k \times l}$, and computes $k$ components each from a $l$-dimensional subspace. With the fixed total latent dimension as 512, we vary $k$ and $l$ choices and plot their effects in Figure 3.

- **IQEs** generally approximate and generalize better (lower test MSE) with larger component size $l$. More components (large $k$) often lead to larger predictions for unreachable pairs in IQE-sum, at the cost of worse approximation with smaller $l$. Small $k$ and large $l$ usually perform well, but the best results are from using a large component size $l$ while still maintaining some number of components (e.g., $k \geq 8$ for IQE-sum).
- **PQE** behavior is largely unaffected by this choice, and generally underperforms IQE except for a few extreme choices. With large $k$, PQE-GG tends to overflow when predicting on unreachable pairs, while PQE-LH does not suffer from this issue.

**Strict quasimetric structure is better than regularizing unconstrained models.** As shown in Table 2, adding a triangle inequality regularizer only has marginal benefits, and is still significantly worse than the strictly enforced quasimetric structure from IQEs.

### 5.2. Random Graphs

Using three randomly generated graphs, we compare different quasimetric models' ability to fit different quasimetric structures with different training set sizes (by regressing graph distances in a similar fashion as above), and to generalize on test pairs. We visualize the distinct structures of three graphs Figure 6 in the appendix.

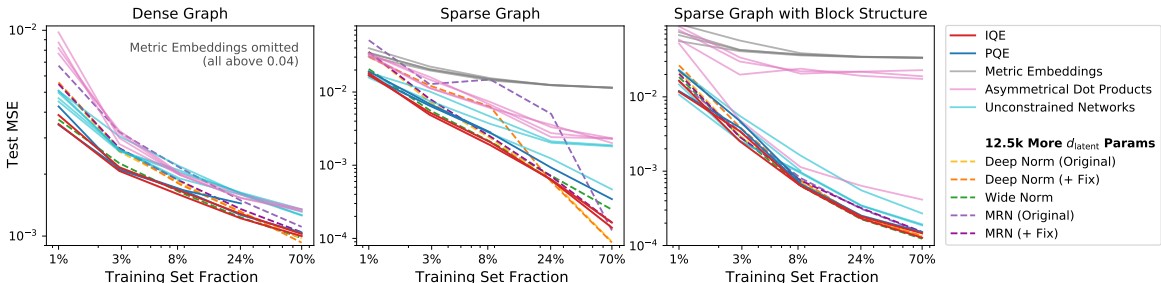

Figure 4: Modeling graphs of different structures. Deep Norm, Wide Norm and MRN use latent quasimetric head with 12,500 more parameters than IQEs ($\leq 1$ parameter) and PQEs. The much simpler IQEs are comparable or better than them, and outperform all other methods.

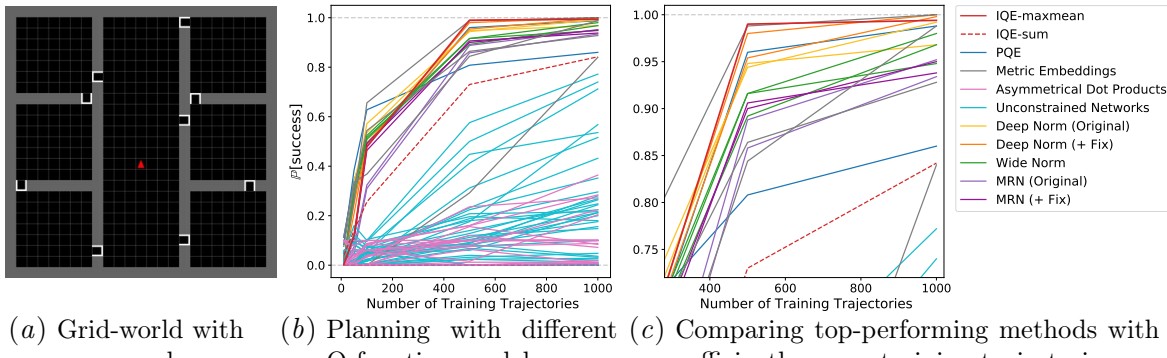

(*a*) Grid-world with one-way doors.

(*b*) Planning with different Q-function models.

(*c*) Comparing top-performing methods with sufficiently many training trajectories.

Figure 5: Offline goal-conditioned Q-learning results on a simple grid-world with four directional actions. Using different goal-conditioned Q-function models leads to different inductive biases and planning success rates. We use one-step greedy planning w.r.t. learned Q-function.

**IQEs are simple, efficient and effective.** In Figure 4, IQEs are consistently among the top performing methods for all three distinct structures, when many other latent quasimetric baselines use 12,500 more parameters. While Deep Norm slightly outperforms IQEs on the sparse graph with a large training set size, IQEs are consistently better with smaller training sets on all graphs, and comparable in other settings.

### 5.3. Offline Q-Learning

In planning, optimal plan costs to reach target states have a quasimetric structure (Bertsekas and Tsitsiklis, 1991; Tian et al., 2020; Wang and Isola, 2022). We use quasimetric models as drop-in replacements for goal-conditioned Q-function models in offline Q-learning on grid-world environment with one-way doors (Figure 5(*a*); Wang and Isola, 2022).

**Quasimetric structure improves sample efficiency of offline RL.** In Figure 5(*b*), quasimetric models generally outperform the widely used unconstrained networks and asymmetrical dot products, which do not have similar geometric constraints. Deep Norm and MRN also benefit from our proposed fixes that strictly enforce quasimetric constraints. Indeed, quasimetric structures greatly improve sample efficiency, and leads to better planning results with fewer training trajectories. As training set size increases, IQE-maxmean out-

performs all other latent quasimetric baselines, and only Deep Norm (with our fix) remains comparable but requires many more parameters (Figure 5($c$)).

**IQE-maxmean is effective in offline RL.** IQE-maxmean performs significantly better than IQE-sum on this task. We suspect that the max operation in the maxmean reduction may encourage the learned function to be more conservative (i.e., outputting larger distances), which improves offline RL (Kumar et al., 2020). Indeed, other methods that also use the maxmean reduction (Deep Norm and Wide Norm) generally perform better than MRN and PQEs, which use a summation reduction. IQE-maxmean outperforms most other such methods that use maxmean reduction, suggesting that its effectiveness is not just due to the reduction, but also the IQE formulation.

## 6. Discussion

In this work, we present four desired criteria when adding quasimetric structures to machine learning models (Section 2). Our proposed Interval Quasimetric Embedding (IQE) is the first method that satisfies all four (Section 3), and has strong theoretical guarantees (Section 3.1) and empirical performance (Section 5). We believe that IQE's simple yet powerful form can enable more machine learning applications of quasimetrics in modeling asymmetrical geometric structures, and that our four criteria are helpful in developing novel and better quasimetric structures.

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

## Appendix A. Deriving IQE From PQE

Here we will derive IQE via modifying the PQE-LH formula to scale linearly with latent (i.e., to have latent positive homogeneity).

Recall the PQE-LH formula from Wang and Isola (2022):

$$d_{\mathsf{PQE\text{-}LH}}(u, v; \alpha) = \sum_i \alpha_i \cdot (1 - \exp(-\sum_j (u_{ij} - v_{ij})^+)). \tag{6}$$

PQE-LH

To make it scale linearly with latents, we must avoid the exponentiation transform on latent vector values, and instead use the latent vector to control a linear quantity. Therefore, we will reformulate the outer sum as an integral, and use latent vector to indicate where the summand (now integrand) has non-zero values.

First, we reformulate Equation (6) with an integration without weighting (by $\alpha$):

$$d_{\mathsf{Integral\text{-}PQE\text{-}LH}}(u, v) = \int_x (1 - \exp(-\sum_j (h_j(u; x) - h_j(v; x))^+)) \, \mathrm{d}x. \tag{7}$$

Integral PQE-LH

PQE-LH is derived by considering processes only activated on sets of the form $[x, \infty)$ (half-lines). Inspired by this choice, we consider $h_j(u; x) = \begin{cases} c & \text{if } x > u_j \\ 0 & \text{otherwise} \end{cases}$, for some $c > 0$.

Then

$$d_{\text{Integral-PQE-LH}}(u, v; c) = \int_x (1 - \exp(-c \cdot |\{j \colon x \in [u_j, \max(u_j, v_j)]\}|)) \, \mathrm{d}x. \tag{8}$$

Take $c \to \infty$, we have

$$d_{\text{Integral-PQE-LH}}(u, v) = \int_x \mathbb{1}(\exists j, x \in [u_j, \max(u_j, v_j)]) \, \mathrm{d}x \tag{9}$$

$$= \left| \bigcup_j [u_j, \max(u_j, v_j)] \right|, \tag{10}$$

which is exactly the IQE component.

Then, for expressivity, we combine several such components and obtain IQEs.

## Appendix B. Proofs

**Proof** [Theorem 2]

- **Proof for IQE-maxmean.**

  At $l = 1$, IQE-maxmean formula can exactly recover the MRN asymmetrical component $d_{\text{asym}}$. By Theorem 2 of Liu et al. (2022), $(f_1, d_{\text{asym}})$ can exactly represent $d$ for some $f_1$. Therefore, the same results apply to $d_{\text{IQE-maxmean}}$.

- **Proof for IQE-sum.**

  For $d_{\text{IQE-sum}}$, we present a novel construction that allows it to represent any quasipartition, and thus any convex combination of quasipartitions. Then, by Lemma C.5 of Wang and Isola (2022), some convex combination of quasipartitions admits a $\mathcal{O}(t \log^2 n)$ embedding.

  WLOG, consider any quasipartition $\pi$ represented as an order embedding $g \colon \mathcal{X} \to [n]^m$. That is,

  $$\pi(u, v) = \begin{cases} 0 & \text{if } g(u) \leq g(v) \text{ coordinate-wise} \\ 1 & \text{otherwise.} \end{cases} \tag{11}$$

  Consider vectors $e_i \in \{0, 1\}^n$, where only the first $i$ dimensions are 0's, and the rest are 1's. These vector nicely connect the IQE component structure (union of intervals) with the order embedding structure (conjunction over coordinate-wise comparisons).

  For any latent $u, v$ and any $i \in [m]$,

  $$\bigcup_{j=1}^n \left[ (e_{g_i(u)})_j, \max((e_{g_i(u)})_j, (e_{g_i(v)})_j) \right] = \begin{cases} \varnothing & \text{if } g_i(u) \leq g_i(v) \\ [0, 1] & \text{otherwise.} \end{cases} \tag{12}$$

Construct mapping

$$f(u) \triangleq [e_{g_1(u)} :: e_{g_2(u)} :: \cdots :: e_{g_m(u)}] \in \{0,1\}^{mn}, \tag{13}$$

where :: denotes concatenation.

Then, for any latent $u, v$,

$$\bigcup_{j=1}^{n} \big[ f_i(u), \max(f_i(u), f_i(v)) \big] = \begin{cases} \varnothing & \text{if } g(u) \le g(v) \text{ coordinate-wise} \\ [0,1] & \text{otherwise.} \end{cases} \tag{14}$$

By using scaled $f$, each IQE component can thus represent arbitrary scaled quasi-partition. Thus IQE-sum can exactly represent any convex of quasipartitions using a polynomial-sized neural encoder.

■

**Proof** [Theorem 3] In proof of Theorem 2, a reduction from MRN asymmetrical part to IQE-maxmean is given. The same reduction can be applied here. Invoking Theorem 2 of Liu et al. (2022) leads to the desired result. ■

**Proof** [Theorem 4] MRN approximation results (same as Theorems 2 and 3) are proved showing that an asymmetric norm (i.e., semi-norm) universally approximate quasimetrics (Theorem 2 of Liu et al., 2022). Deep Norm and Wide Norm can approximate any semi-norm (Theorem 2 of Pitis et al., 2020) and thus have the same properties. ■

## Appendix C. Fixes for Deep Norm and MRN

The original formulations of Deep Norm and MRN actually do not fully satisfy quasimetric constraints. Here we highlight where they are wrong and explain our proposed fixes. In Section 5, we compare with both the original and the fixed version.

### C.1. Deep Norm May Be Negative

In the original work (Pitis et al., 2020), Deep Norm is formulated as a combinations over several components, each of which is the output of a maxrelu activation, where

$$\text{maxrelu}(x, y; \alpha, \beta) \triangleq [\max(x, y), \alpha \cdot \text{ReLU}(x) + \beta \cdot \text{ReLU}(y)]. \tag{15}$$

However, the max component is not guaranteed to be non-negative, so the eventual output may be negative, and Deep Norm may not be a valid quasimetric.

To fix this issue, we simply replace the final activation to be simply ReLU. As shown in Table 2 and Figure 5(c), this fix improves performance.

### C.2. MRN May Not Be A Quasimetric

In the original work (Liu et al., 2022), MRN is formulated as the sum of a symmetrical component and an asymmetrical component. While the asymmetrical component $\max_i (h(u)_i - h(v)_i)^+$ is a valid quasimetric, the symmetrical component $\|\phi(u) - \phi(v)\|_2^2$ is not a metric.

To fix this issue, we simply remove the square and use $\|\phi(u) - \phi(v)\|_2$ as the symmetrical component. As shown in Table 2 and Figure 5(c), this fix improves performance.

## Appendix D. Experiment Details

Across all three tasks, our architecture choices and optimization settings generally follow the prior work (Wang and Isola, 2022). For completeness, we report the full details below. We run each experiment setting for 5 runs with different seeds, and present the aggregated results.

### D.1. Large-Scale Social Graph

**Architecture.** For all embedding methods (i.e., asymmetrical dot products and latent quasimetrics), we use a 128-2048-2048-2048-512 ReLU encoder with Batch Normalization (Ioffe and Szegedy, 2015) after each activation. The encoders take in 128-dimensional inputs and output 512-dimensional latent vectors. Unconstrained networks use a similar 256-2048-2048-2048-512-1 ReLU network, mapping concatenated the 256-dimensional input to a scalar output.

**Optimization.** We use 80 training epochs, batch size 1024, and the Adam optimizer (Kingma and Ba, 2014), with learning rate decaying from $10^{-4}$ to 0 by the cosine schedule without restarting (Loshchilov and Hutter, 2016). The training objective is MSE on the $\gamma$-discounted distances, with $\gamma = 0.9$. When applying the triangle inequality regularizer (for asymmetrical dot products and unconstrained networks), $342 \approx 1024/3$ triplets are uniformly sampled at each iteration to compute the regularizer term.

**Hyperparameters.** For the following baselines, we tune their hyperparameters:

- **IQE** and **PQE**: component size $l \in \{8, 16, 32, 64\}$ (and thus correspondingly number of components $k \in \{64, 32, 16, 8\}$).

- **Deep Norm (both the original version and the version with our fix)**: three layers with hidden size $\in \{128, 512\}$, where final number of output components equals the hidden size.

- **Wide Norm**: 32 components each with size $\in \{32, 48, 128\}$.

- **MRN (both the original version and the version with our fix)**: Both the symmetrical and the asymmetrical projection heads have two layers with hidden size $\in \{128, 512\}$, where the projector output dimension equals the hidden size.

- **Asymmetrical Dot Products and Unconstrained Networks with $\Delta$-inequality regularizer**: regularizer weight $\in \{0.3, 1, 3\}$.

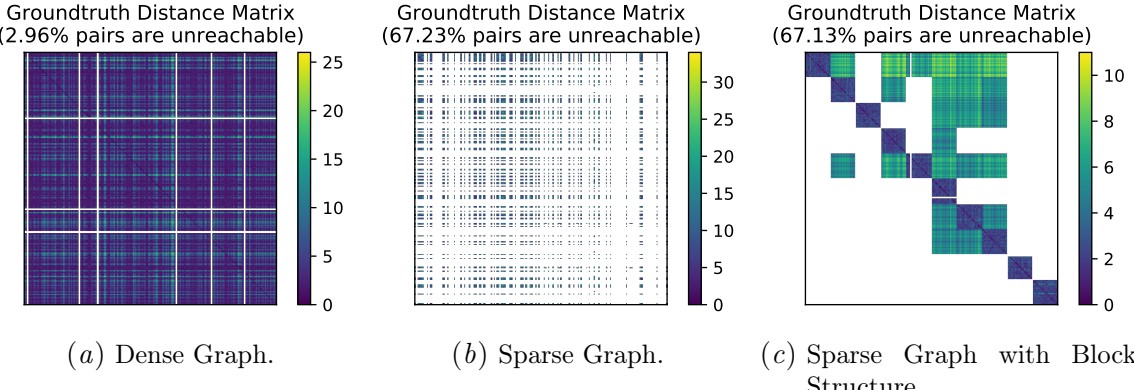

(a) Dense Graph.     (b) Sparse Graph.     (c) Sparse Graph with Block Structure.

Figure 6: Structures of random graphs used in Section 5.2 experiments.

## D.2. Random Graphs

**Graph Structures and Datasets.** We use the same randomly generated 300-node graphs and train-validation splits as Wang and Isola (2022). See Figure 6 for a visualization on their structures. We test the models on 5 different training set ratios, where training set fraction (of the total 90,000 pairs) are evenly spaced on the logarithm scale from 0.01 to 0.7. We use the same 64-dimensional node features from Wang and Isola (2022).

**Architecture.** For all embedding methods (i.e., asymmetrical dot products and latent quasimetrics), we use a 4-128-128-128-48 ReLU encoder, mapping to 48-dimensional latent vectors. Unconstrained networks use a similar 128-128-128-128-48-1ReLU network, mapping concatenated the 128-dimensional input to a scalar output.

**Optimization.** We use 3000 training epochs, batch size 2048, and the Adam optimizer (Kingma and Ba, 2014), with learning rate decaying to 0 by the cosine schedule without restarting (Loshchilov and Hutter, 2016). All models are optimized w.r.t. MSE on the $\gamma$-discounted distances, with $\gamma = 0.9$. When running with the triangle inequality regularizer, $342 \approx 1024/3$ triplets are uniformly sampled at each iteration.

**Hyperparameters.** For all methods, learning rates are tuned among $\{10^{-4}, 3 \times 10^{-4}, 10^{-3}, 3 \times 10^{-3}, 10^{-2}\}$. We run with the following hyperparameters for each method:

- **IQE** and **PQE**: component size $l \in \{4, 6, 8, 12\}$ (and thus correspondingly number of components $k \in \{12, 8, 6, 4\}$).

- **Deep Norm (both the original version and the version with our fix)**: three layers with 48 hidden size, where final number of output components equals the hidden size.

- **Wide Norm**: 12 components each with size 11, 22 components each with size 6, or 6 components each with size 22.

- **MRN (both the original version and the version with our fix)**: Both the symmetrical and the asymmetrical projection heads have two layers with 58 hidden size, where the projector output dimension equals the hidden size.

- **Asymmetrical Dot Products and Unconstrained Networks with $\Delta$-inequality regularizer**: regularizer weight $\in \{0.3, 1, 3\}$.

**Figure 4 Details.** For clarity, we didn't plot all settings for each family of method. Instead, we plot only the best member within the following families: IQE-sum, IQE-maxmean, PQE-LH, PQE-GG, Wide Norm, Deep Norm (original), Deep Norm (with fix), MRN (original), MRN (with fix), unconstrained networks with each particular output parametrization (i.e., we tune triangle inequality regularizer weight for unconstrained networks that output discounted distances, and only plot the best), asymmetrical dot products with each particular output parametrization (done in an identical way with unconstrained networks). All metric embeddings are plotted.

### D.3. Offline Q-Learning

**Environment and Datasets.** Following Wang and Isola (2022), we use the grid-world environment based on `gym-minigrid` (Chevalier-Boisvert et al., 2018), and use a training dataset of trajectories, collected by an $\epsilon$-greedy planner with groundtruth quasimetrics, with a large $\epsilon = 0.6$, where each trajectory is capped at 200 steps.

**Algorithm.** Following Wang and Isola (2022), we use a modified version of MBOLD (Tian et al., 2020). Please refer to Wang and Isola (2022) for details on the modifications. To use encoder-based methods, we train them with goals as state-action pairs. In evaluation, for current state $s$, candidate action $a$ and a given goal state $g$, we use $\frac{1}{|\mathcal{A}|} d((s, a), (g, a')) - 1$ as the predicted distance from $(s, a)$ to $g$. For unconstrained networks, we also test the original formulation where goals are simply states.

**Architecture.** For all embedding methods (i.e., asymmetrical dot products and latent quasimetrics), we use a 18-2048-2048-2048-1024 ReLU encoder with Batch Normalization (Ioffe and Szegedy, 2015) after each activation. The encoders take in 18-dimensional states and output four 256-dimensional latent vectors, one for each actions. For unconstrained networks,

- With the new formulation using state-action pairs as goals, we use a similar 36-2048-2048-2048-256-16 network to map input state pairs to a value for each $|\mathcal{A}| \times |\mathcal{A}|$ action pair options;

- With the original formulation using states as goals, we use a similar 36-2048-2048-2048-256-4 network to map input state pairs to a value for each action.

**Optimization.** We use 100 training epochs, batch size 1024, and the Adam optimizer (Kingma and Ba, 2014), with learning rate decaying from $10^{-4}$ to 0 by the cosine schedule without restarting (Loshchilov and Hutter, 2016). The training objective is the same as usual Q-learning: MSE on the $\gamma$-discounted distances, with $\gamma = 0.95$. When applying the triangle inequality regularizer (for asymmetrical dot products and unconstrained networks), $341 \approx 1024/3$ triplets are uniformly sampled at each iteration to compute the regularizer term.

**Planning.** We perform simple greedy 1-step planning without any lookahead. At each step, we query the learned Q-function for all action choices, and select the best action. In evaluation, we plan for 50 goals, and cap trajectory length at 300 steps.

**Hyperparameters.** For each method, we run the following hyperparameter choices:

- **IQE**: component size $l = 8$ (and thus correspondingly number of components $k = 16$), selected based on effects of $(k, l)$ discussed in Section 5.1.
- **PQE**: component size $l = 4$ (and thus correspondingly number of components $k = 32$), following Wang and Isola (2022).
- **Deep Norm (both the original version and the version with our fix)**: three layers with hidden size $\in \{64, 128\}$, where final number of output components equals the hidden size.
- **Wide Norm**: 32 components each with size $\in \{32, 48, 128\}$.
- **MRN (both the original version and the version with our fix)**: Both the symmetrical and the asymmetrical projection heads have two layers with hidden size $\in \{128, 512\}$, where the projector output dimension equals the hidden size.
- **Asymmetrical Dot Products and Unconstrained Networks with $\Delta$-inequality regularizer**: regularizer weight $\in \{0.3, 1, 3\}$.

Each choice is plotted as a line in Figure 5.

