# OpenReview forum: "Improved Representation of Asymmetrical Distances with Interval Quasimetric Embeddings"
_NeurIPS.cc/2022/Workshop/NeurReps — NeurReps 2022 Poster_

### Official Review · Reviewer_ViUa · 2022-10-12
**Review of Improved Representation of Asymmetrical Distances with Interval Quasimetric Embeddings**

**Confidence:** 2
**Soundness:** 3
**Presentation:** 3
**Contribution:** 3
**Overall Rating:** 6

**Summary:**

The authors present directed distance metrics which fulfill four proscribed desiderata. They demonstrate the metrics are universal, and go on to show experimentally that embeddings equipped with this metric (obtained using encoders) are more predictive than competing approaches.

**Questions:**

Questions in strengths and weaknesses

**Limitations:**

Comments in strengths and weaknesses.

**Recommended Decision:**

2: Borderline

**Relevance:**

3: Solid fit

**Strengths And Weaknesses:**

Strengths:
- The paper is written in an energetic style
- The experiments are extensive
- The idea of the metric is very original
- The research question is important
- Theorem 2 is interesting

Weaknesses
- The paper is hard to follow, because it assumes familiarity with a lot of non-generalist terminology. For example, do the authors define the gamma-discounted distance? Why do the authors describe u, v as tensors rather than matrices? I appreciate that the latter is an example of the former, but would the word matrix not be more widely understood? What does "head" mean, in "Latent Quasimetric Head #parameters"?
- I do not find the four stated desiderata entirely 'self-evident'; if homogeneity in scale is desirable, why not, say, invariance to translation? Also, would it not be desirable for the embedding dimension to be low, for the metric to be continuous w.r.t. Euclidean metric, etc? (Note I can only guess at what "head" means in the third desideratum, so I am not sure if I agree with it or not.)
- The writing is a very informal at times; e.g. "a lot" in the table in page 3
- While I see that the experiments are encouraging, they are quite complex and rely on several other analysis techniques which are not themselves the most transparent (e.g. node2vec). As a result, it is hard to know how much this proposed distance says anything about any true notion of directed distance in the data - which is the goal in various scientific problems (such as causal inference) alluded to in the introduction. If I understand what gamma-discounting is (I can't be sure, because I have not been able to find a definition in the paper), then does it reward getting short distances right and worry less about long distances? Is rewarding local over global fidelity always desirable? For theory, downstream procedure development, data analysis, this distance function would seem extremely complex and irregular compared to a dot product (even if it is not a quasi-metric). This puts a very high bar on what sort of performance improvement adopting this distance would have to bring, which I am not sure is demonstrated to have been met.


**Submission Track:**

Proceedings Paper (9 Page)

---

### Official Review · Reviewer_MYgc · 2022-10-15
**Good arguments for explicit, parameter-free constraint enforcement in quasimetric learning**

**Confidence:** 4
**Soundness:** 4
**Presentation:** 3
**Contribution:** 3
**Overall Rating:** 7

**Summary:**

A new model is introduced in the class of "latent quasimetrics", i.e., approximations for quasimetrics via the composition of unconstrained latent embeddings with binary real-valued functions (approximately) satisfying the properties of a quasimetric. The main contribution is a modification of the Poisson Quasimetric Embedding (PQE) model, homogenising it and fully removing its free parameters, while strictly satisfying the quasimetric properties. The authors suggest that this leads to better generalisation capabilities, and the claim is substantiated by universal approximation results and empirical experiments on supervised learning of weighted digraphs and Q-functions.

**Questions:**

A more careful explanation would be desirable for the reasons motivating the criteria on parametrisation and homogeneity. In particular, the structural relationship between the latent embedding and the subsequent quasimetric is only briefly established in the text. The interpretation of results, both theoretical and empirical, would be improved by a notation which consistently displays and distinguishes the parametrisations of the latent embeddings and (approximate quasi-)metrics. And while documenting model behaviour solely in terms of test loss is standard practice in contemporary ML, the work would become significantly more interesting if the effects of the explicitly enforced quasimetric structure on the learned latent representations were described or visualised somehow.

**Limitations:**

It would be worthwhile to describe more precisely the mathematical sense in which this work is "adding quasimetric structures to ML models", i.e., pointing out the Euclidean input and real-valued output, which could invite readers to think about more general uses of similar models, e.g., discrete domains or algebraic constraints in the input, or other "generalised metric spaces" in the output.

**Recommended Decision:**

3: Accept

**Relevance:**

3: Solid fit

**Strengths And Weaknesses:**

The suggested explicit quasimetric constraint is novel, and its relationship to the precursor PQE and the broader related ML literature is clearly described. The text is easy to follow, and the results are intriguing. I expect the proposed method to be of practical interest to problems involving quasimetrics, and its simplicity might encourage careful thinking about metric structures in the community more generally. While the technical quality and contextualisation is good for ML standards, the work might have stronger impact and the model structure might be more convincing if appropriate references to the mathematical communities of order theory, complex network theory and applied category theory were included.

**Submission Track:**

Proceedings Paper (9 Page)

---

### Official Review · Reviewer_ZFfb · 2022-10-17

**Confidence:** 3
**Soundness:** 3
**Presentation:** 3
**Contribution:** 3
**Overall Rating:** 7

**Summary:**

This work proposes interval quasimetric embeddings - which satisfies the metric constraints apart from symmetry property - while satisfyig the universal approximation property, and having latent postive homogenity (which other prior methods don't possess) and thereby allow to be better utilized in downstream tasks. The authors show improved performance  (in terms of MSE and $l_1$ error in comparison to prior works on the Berkeley-Stanford Web graph, on random graphs and a task on offline RL.

**Questions:**

Please address the concerns raised in the weakness sections.

**Limitations:**

The impact on the field of neuroscience is unclear.

**Recommended Decision:**

3: Accept

**Relevance:**

2: Limited relevance

**Strengths And Weaknesses:**

**Strengths:**
1. The paper is well written and easy to read.
2. The theory is well structured and the proofs appear correct.
3. The experiments corroborate the theory and strengthen the proposed IQE over prior works.

**Weaknesses:**
1. The general impact of a lower 'predicted distance' in comparison to PQE-GG, MRN - is not provided
2. The direct application to neuroscience is not clear - i.e. a purely geometric paper. Would be better suited to the conference if the authors can add to this.


**Submission Track:**

Proceedings Paper (9 Page)

---

### Decision · Program_Chairs · 2022-10-21

Accept (Poster)